# Ezetimibe-Loaded Nanostructured Lipid Carrier Based Formulation Ameliorates Hyperlipidaemia in an Experimental Model of High Fat Diet

**DOI:** 10.3390/molecules26051485

**Published:** 2021-03-09

**Authors:** Yogeeta O. Agrawal, Umesh B. Mahajan, Vinit V. Agnihotri, Mayur S. Nilange, Hitendra S. Mahajan, Charu Sharma, Shreesh Ojha, Chandragouda R. Patil, Sameer N. Goyal

**Affiliations:** 1Department of Pharmaceutics and Quality assurance, R. C. Patel Institute of Pharmaceutical Education and Research, Shirpur, Dhule, Maharashtra 425405, India; vvagnihotri@rcpatelpharmacy.co.in (V.V.A.); goyalyogeeta@gmail.com (M.S.N.); hsmahajan@rcpatelpharmacy.co.in (H.S.M.); 2Department of Pharmacology, R. C. Patel Institute of Pharmaceutical Education and Research, Shirpur, Dhule, Maharashtra 425405, India; ubmahajan@rcpatelpharmacy.co.in (U.B.M.); crpatil@rcpatelpharmacy.co.in (C.R.P.); sameer.goyal@svkm.ac.in (S.N.G.); 3Department of Internal Medicine, College of Medicine and Health Sciences, United Arab Emirates University, Al-Ain P.O. Box 15551, United Arab Emirates; charusharma@uaeu.ac.ae; 4Department of Pharmacology and Therapeutics, College of Medicine and Health Sciences, United Arab Emirates University, Al-Ain P.O. Box 15551, United Arab Emirates; shreeshojha@uaeu.ac.ae; 5SVKM’s, Institute of Pharmacy, Dhule, Maharashtra 424001, India

**Keywords:** ezetimibe, NLCs, hyperlipidemia, high fat diet, lipid profile

## Abstract

Ezetimibe (EZE) possesses low aqueous solubility and poor bioavailability and in addition, its extensive hepatic metabolism supports the notion of developing a novel carrier system for EZE. Ezetimibe was encapsulated into nanostructured lipid carriers (EZE-NLCs) via a high pressure homogenization technique (HPH). A three factor, two level (2^3^) full factorial design was employed to study the effect of amount of poloxamer 188 (X1), pressure of HPH (X2) and number of HPH cycle (X3) on dependent variables. Particle size, polydispersity index (PDI), % entrapment efficiency (%EE), zeta potential, drug content and in-vitro drug release were evaluated. The optimized formulation displays pragmatic inferences associated with particle size of 134.5 nm; polydispersity index (PDI) of 0.244 ± 0.03; zeta potential of −28.1 ± 0.3 mV; % EE of 91.32 ± 1.8% and % CDR at 24-h of 97.11%. No interaction was observed after X-ray diffraction (XRD) and differential scanning calorimetry (DSC) studies. EZE-NLCs (6 mg/kg/day p.o.) were evaluated in the high fat diet fed rats induced hyperlipidemia in comparison with EZE (10 mg/kg/day p.o.). Triglyceride, HDL-c, LDL-c and cholesterol were significantly normalized and histopathological evaluation showed normal structure and architecture of the hepatocytes. The results demonstrated the superiority of EZE-NLCs in regard to bioavailability enhancement, dose reduction and dose-dependent side effects.

## 1. Introduction

Hypercholesterolemia (hyperlipidemia) affects an enormous population. This condition is determined with a decline high density lipoprotein (HDL) levels (cholesterol < 40 mg/dL) and increased levels of low density lipoprotein (LDL) (cholesterol >1 90 mg/dL) along with triglycerides (TG) in plasma (>200 mg/dL), which eventually results in the progression to atherosclerosis [1]. Excessive cholesterol, however, puts a person at risk of developing heart disease [2]. Hypercholesterolemic patients, because of excess cholesterol in the blood stream, may lead to excessive deposition of cholesterol on the walls of coronary arteries which supply blood to the heart (coronary arteries) [3].

Ezetimibe is a cholesterol absorption inhibitor drug used for the treatment of primary hypercholesterolemia. Ezetimibe decreases raised levels of high total cholesterol (total-C), low density lipoprotein cholesterol (LDL-C), apolipoprotein B (Apo B), and non-high density lipoprotein cholesterol (non HDL-C) in patients with hyperlipidemia [4]. This BCS class II therapeutic moiety is an azetidine derivative with a low water-solubility profile and a high permeability index (log P of 4.56). Once absorbed into the portal venous system, EZE undergoes extensive first-pass metabolism and the rate of this first-pass hepatic uptake inversely relates to the systemic bioavailability [5]. EZE is primarily used to reduce the intestinal cholesterol levels while in the alkaline environment of the intestine, its absorption is limited. Thus, slow and limited drug release occurs under alkaline conditions on, often manifested as low oral bioavailability. Furthermore, bioavailability is also affected by high lipophilicity, pre-systemic clearance in the gastrointestinal mucosa and P-gp efflux mechanism. In this context, novel approaches that improve the dissolution and/or apparent solubility are needed for better therapy, especially for poorly water-soluble drugs like EZE [6].

Lipid-based drug delivery scaffolds are potential drug carriers owing to their propensity for solubility improvement of poorly water-soluble and/or lipophilic drugs which ultimately augments the oral bioavailability [7]. Nanostructured lipid carriers (NLC) have been investigated to a great extent to overcome the elementary drawbacks involved with other nanocarriers [8,9,10]. Wherefore we could regard NLCs as new-generation derivatives of the solid lipid nanoparticle (SLN) family which are mainly composed of a solid lipid matrix entrapping different incompatible liquid lipid nano-antechambers mostly stabilized by surfactants [11]. The imperfect crystal structure in NLC formulations hold the encapsulated drugs purged from nano-compartments thereby increasing the drug loading. Additionally, NLCs are biocompatible and offer convenience in scaling during production and show prolonged drug release profiles [12]. Once NLC formulations reach the intestinal lumen, they are however vulnerable to enzymatic digestion by lipases and co-lipases), which transform them into readily absorbed mixed micelles [13]. Furthermore, the drug transport via a lymphatic route or Peyer’s patches synergistically results in improved oral absorption of drugs encapsulated in NLCs [14,15].

Advanced therapies using novel nanotechnology strategies are the need of the hour with the better understanding of hypercholesterolemia and related pathologies. A number of formulation approaches for the development of efficacious medications and safe anti-hyperlipidemic drugs are in progress. Due to the inherent problems of this potent drug various formulations have already been reported across the world, like nanocrystals, cyclodextrin inclusion complexes, suspensions, and solid lipid nanoparticles. However, they exhibit limitations of poor drug encapsulation efficiency, expulsion of the drug during storage and high-water content in the formulations [16].

Ezetimibe has poor aqueous solubility (0.00846 gm/L) and low bioavailability (35%). It also shows limited drug absorption in the alkaline environment of the intestine. Therefore, the objective of this work was to improve the dissolution and in turn the solubility to enhance its oral bioavailability. Numerous previous studies on this topic were reported. Gaba et al. and Zhou et al., successfully studied oral bioavailability enhancement via nanostructure lipid carriers [17,18]. Prajapati et al., described that the objective of preparing NLCs was to avoid its first-pass metabolism by way of lymphatic uptake [19]. Also, we assume that the efficiency of ezetimibe could be improved by loading the drug in nanostructured lipid carriers as previously reported by Khan et al. [20]. In our previous studies (Agrawal et al. [21]) we also emphasized that by using nanostructured lipid carriers the efficiency and bioavailability of drugs could be improved significantly and it also helps to avoid first pass metabolism. Several attempts have been made to enhance the bioavailability of ezetimibe via the oral route. A nanoemulsion has been developed by Bali et al., which was not suitable due to physical stability limitations at room temperature [22]. Ha et al. tried to enhance the dissolution and bioavailability of ezetimibe by using amorphous solid dispersion nanoparticles [23]. However, the major concern in such formulation is the solute concentration to control the particle size in the bulk. Thadkala, et al. tried to integrate EZE through a nanosuspension [24], which suffered from the major drawback of crystal growth and stability. In addition, this formulation would not help in preventing lympthatic uptake and protect a BCS class II drug like EZE which undergoes first pass hepatic metabolism. Solid lipid nanoparticles were successfully employed as a carrier for EZE by Din, et al. [6], but SLN per se has a potential pitfall of a gelation tendency and lipid particle growth. Furthermore, it also increases the risk of expulsion of the drug from the formulation upon storage due to polymorphic transitions. A recently published review by Haider et al., presented numerous advantages of NLCs in relation to safety, efficacy and pharmacokinetic properties [25]. Das et. al. reported the formulation and comparative evaluation of SLN and NLC, where NLC was proven to have a better stability profile than SLN [26]. Ghasemiyeh et al. explained that the shortcomings of SLNs can be overcome by development of NLCs [27].

The lack of comprehensive studies on the development of ezetimibe NLCs via optimization, including in vitro and in vivo analysis, are the foundation for this investigation.

NLCs are newer and novel technological innovations with massive possibilities of oral bioavailability enhancement of lipophilic drugs. These nanosystems protect drugs from gastric degradation by luminal enzymes. NLCs for the most part are composed of lipids, which favours bioavailability enhancement along with improvement in the solubility profile [28], so the objective of the study was to explore the feasibility of encapsulating ezetimibe in NLCs to allow sustained release of the drug by improving its dissolution rate and oral bioavailability. In this regard, an ezetimibe-loaded NLC formulation was prepared by a high speed homogenization technique, optimized by employing a three factor, two level (2^3^) full factorial design, and extensively characterized by Fourier transform infrared spectroscopy (FTIR), differential scanning calorimetry (DSC), powder X-ray diffraction (PXRD), transmission electron microscopy (TEM), etc. Further, drug particle size, zeta potential, entrapment efficiency, stability study, in-vitro and in-vivo studies were evaluated in an experimental model of high fat diet induced hyperlipidemia in Wistar rats.

## 2. Results and Discussion

In the process of formulation development, optimization of the method with respect to quantities of ingredients and process parameter settings plays a vital role in order to obtain the desired characteristics of the product and also to ensure smooth transfer of the experimental batches to large scale processing. Statistical models are very useful as they assist formulation scientists with an understanding of the theoretical formulation and target processing parameters such as the range of excipient concentrations to be employed in the formulation. Optimization techniques provide both in-depth understanding and an ability to explore and define ranges of formulation and processing factors. Thus, optimization may be defined as the process of identifying the experimental conditions which lead to the best value of the desired response.

### 2.1. Phase Solubility Study of EZE between the Lipids

Based on the screening of liquid lipids for the maximum solubility of the drug, Capmul PG 8 was identified as the most suitable liquid lipid to be used in the EZE-NLCs. In the case of solid lipid screening glycerol monostearate (GMS) emerged as the most suited solid lipid for the formulation of EZE-NLCs. Moreover, both the lipids are non-toxic in nature, have approved regulatory status (GRAS), and are inexpensive and hence were used in the formulation.

### 2.2. Development and Optimization of EZE Loaded NLCs by Factorial Design

The EZE-NLC dispersions were prepared by using a hot high-pressure homogenization technique. For the preparation of EZE-NLCs, GMS was selected as a solid matrix and Capmul PG 8 as liquid lipid on the basis of phase solubility studies at a fixed concentration of 70:30 [29]. Depending upon the results obtained from trial batches and referred values, the optimal value range of the variables was surfactant concentration (A) of 0.5–1 gm, pressure of HPH (B) of 500–700 bar and number of cycles in HPH (C) from 5–7.

Table 1 shows the 2^3^ full factorial design with concentration of poloxamer 188 (factor A), pressure of HPH (factor B) and number of HPH cycles (factor C) as independent variables while particle size and % entrapment efficiency are the responses. The results of the experimental design were observed using the Design-Expert^®^ software. Each response coefficient was studied for its statistical significance at 90 % confidence level. A p value of ‘Prob > F’ less than 0.05 indicates that model terms are significant and values greater than 0.05 indicates that model terms are insignificant and should be removed from the analysis to generate the reduced model.

### 2.3. Optimization Data Analysis and Model Validation

#### 2.3.1. Fitting of the Data to the Model

The ranges of responses Y1 and Y2 were 134.5 to 614.4 nm and 80.11 to 91.32%, respectively. All the responses observed for the eight formulations prepared were fitted to various models using the Design- Expert^®^ software. It was observed that the best fitted models were main effect for the particle size and entrapment efficiency (EE%). The values of *R*^2^, adjusted *R*^2^, predicted *R*^2^, SD, % CV and mean are given in Table 2, along with the regression equation generated for each response. The ANOVA results for the dependent variables demonstrate that the model was significant for three of the response variables.

#### 2.3.2. Effects of Independent Variables on Particle Size (Y1)

The polynomial equation for PS is as follows:(1)$$PS\left(nm\right)=+314.81+116.03A-10.07B-85.10C+17.45AB-66.32AC-21.37BC$$

The ANOVA results indicate that all the three independent factors and their interactions were found to significantly affect the particle size (PS) (*p*-value < 0.05). The *R*^2^ value (0.9996) indicates linearity. A decrease in the concentration of Poloxamer 188 decreases the PS, which might be due to a concomitant reduction in the interfacial tension and subsequently the formation of a stable NLC suspension. Increases in the pressure of HPH and number of HPH cycles led to a very significant decrease in the particle size of the NLC formulation (Figure 1). Insufficiency of surfactant with 0.1% concentration may explaon to formation of NLCs with large particle size and pre-emulsion with higher globule size [30]. Reduction in PS of EZE-NLCs and pre-emulsion with smaller size globules was observed with 0.5% concentration which might indicate effective reduction in the interfacial tension between the aqueous and the lipid phase. In addition, the surfactant helps to stabilize the newly generated surface and prevent particle aggregation. A remarkable increase in PDI was also observed, which supports the aggregation phenomenon explanation. An increase in the viscosity of the aqueous phase causes decrease in size reduction with sonication [31].

#### 2.3.3. Effect of Independent Variables on Drug Entrapment Efficiency (Y2)

The polynomial equation for EE is as follows:(2)$$EE\left(\%\right)=+85.68-2.05A+1.10B+2.49C-1.06AB+0.99AC+2.500E-003BC$$

As evidenced by the strong positive regression coefficient *R*^2^ value (0.9975) indicating linearity and as per the ANOVA data all three independent factors and their interactions were found to significantly affect the entrapment efficiency (*p* < 0.05). The surfactant concentration had an inverse relationship with the percent drug entrapment. Decreases in the concentration of Poloxamer 188 increased the entrapment efficiency and hence stabilized the NLC (Figure 2). Higher surfactant concentration would form micelles in the water phase during the pre-emulsion stage and redistribute the drug from the NLC to the micelles (negative effect). This means that as amount of surfactant increases, it will tend to increase the solubility of the drug in the aqueous phase and drug is therefore not available for encapsulation in the lipid phase. High pressure would facilitate the increased assimilation of the drug inside lipid nanoparticles by increasing the space for accommodation of the drug in lattice imperfections and thereby stabilizing the formulation. Higher solubility of ezetimibe in Capmul PG8 and increasing number of cycles may also be attributed to structural imperfections resulting in higher drug entrapment in the lipid matrix owing to the increased liquid lipid amount.

Independent variables sequentially affect the particle size and entrapment efficiency. When the concentration of surfactant is low, all the surfactants are engaged in reducing the surface tension, leading to a smaller particle size, larger surface area and more drug adhesion on as a condition for a positive effect. The solution recommended by the Design Expert^®^ software indicated that the optimized EZE-NLCs with high EE and least PS were obtained when the formulation contains 0.5% of Poloxamer188, 700 bars of pressure of HPH and seven cycles of HPH. The observed responses of mean particle size (134.5 nm) and EE (91.32%), presented by the optimized EZE-NLCs formulation, and the predicted value of mean particle size (131.11 nm) and EE (91.39%), generated by the Design Expert^®^ software, were found to be in good agreement. This finding demonstrates that the optimized formulation was reliable in predicting the response of the EZE-NLCs system.

### 2.4. Particle Size and Zeta Potential of Optimized EZE Loaded NLC

Immediately after preparation, optimized EZE-loaded NLC dispersion showed an average particle size of 134.5 ± 2.5 nm and zeta potential (mV) of –28.1 ± 0.3 (Nano ZS90, Malvern Instruments Pvt. Ltd., Malvern, Worcestershire, England, UK) as shown in Figure 3. This is confirmed by previous reported studies to be the suitable size range and zeta potential with respect to bioavailability enhancement, lymphatic transport and stability. The lipid nanoparticles with optimum stabilizer composition are usually stable for prolonged periods. The ζ value of –30 mV to –60 mV is considered optimum. This study demonstrated NLCs with suitable PDI and zeta potential owing to the sufficient thickness of the diffusive layer to avoid particle agglomeration.

### 2.5. Thermal Analysis by Differential Scanning Calorimetry (DSC)

DSC (Mettler Toledo, Columbus, OH, USA) thermograms of bulk EZE, EZE and GMS, bulk GMS and lyophilized EZE-NLCs are shown in Figure 4. EZE showed a characteristic sharp endothermic peak at 163.23 °C (B) and the DSC thermograms of the GMS showed an endothermic peak at 59.14 °C (D). The lyophilized EZE-NLCs showed a very small and not sharp endothermic peak around 146.49 °C (C). This small and not sharp endothermic peak indicates that most of the EZE molecules were converted from a crystalline to the amorphous state, i.e., the maximum entrapment of EZE is in the lipid matrix. It was also concluded that EZE is molecularly dispersed in the lipid matrix, indicating its reduction in crystallinity as the peak intensity of the drug was found to be reduced as shown in Figure 4. It was also reported earlier that the shift of the peak and reduction in peak height whether in lipid or drug can be attributed to the matrix which was composed of the mixture of lipids [32,33].

### 2.6. Morphological Analysis by Transmission Electron Microscopy

TEM reveals that particles were uniformly distributed and have a spherical shape morphology with well-defined boundaries. Moreover, the narrow particle size distribution which is in accordance with polydispersion analysis by Zetasizer invalidates some heterogeneity that could be found in the TEM micrographs. At certain locations particles were positioned very close and one above other appearing to fuse at the surfaces (Figure 5). According to work done by Proetto et al., particles at some locations may begin to coalesce over time, owing to electron-beam induced impairment [32]. Additionally, Torras et al., stated that the loss of discrete particles might be attributed to higher temperatures [33]. We assume, this could be due to the slight melding of particle surfaces, for instance during the cooling phase of the nanoemulsion.

### 2.7. X-ray Diffraction Studies

From the XRD data it was clear that pure ezetimibe showed a highly crystalline nature with peaks at 2 θ values of 8.054, 15.980, 16.615, 17.390, 18.838, 19.556, 20.396, 21.974, 23.710, and 25.539 (Figure 6A). In the physical mixture, a few peaks for ezetimibe and GMS were apparent in Figure 6B, but the sharpness of the ezetimibe peaks was poor as compared to Figure 6A. An evidence of absolutely no peak for EZE reveals the formation of an amorphous molecular dispersion in the EZE-NLC formulation (Figure 6C). These might be explained by the integration of the EZE among the parts of the crystal lattice of the lipid matrix, which builds bonus and substantial lattice defects resulting in surplus imperfections in the crystals. This creates sufficient space in the lipid matrix to hold molecules of EZE during the storage period. The width and intensity of the peaks are in accordance with the DSC results. In general, Souto et al., stated that the resolution of the peaks depends on several aspects like the particle size, the amount of sample used, etc. [34].

### 2.8. In-Vitro Drug Release of EZE-NLCs

The in-vitro release profile of EZE was evaluated for 24 h. Almost 30% of drug was discharged from the NLC during the first two hours, which may be due to the outer drug- enriched lipid matrix, then release began to decelerate and was sustained for 24 h. The NLCs approximately took 24 h for 80% of EZE release, which serves to accentuate the sustained effect achieved through the delivery system. Therefore, we emphasize that a prolonged EZE release was observed from our designed NLC. In order to investigate the drug release mechanism, the release data were fitted to models representing zero order, first-order and Higuchi’s square root of time kinetics. On examining the coefficient of determination values for drug release in the various models (Table 3) i.e., zero order (*R*^2^ = 0.817), first order (*R*^2^ = 0.684) and the Higuchi model (*R*^2^ = 0.8465), the higher *R*^2^ values (*R*^2^ ≤ 1) indicated that drug release from the EZE-NLCs formulation followed a diffusion controlled mechanism (Higuchi model, Figure 7). Hence the drug release from NLC showed an early rapid release phase followed by a sustained release period.

A more stringent test was used to distinguish between the mechanisms of drug release. The release data was fitted to the Peppas exponential model, Mt/M∞ = Ktn, where Mt/M∞ is the fraction of drug released after time t; k is the kinetic constant; and n is the release exponent which characterizes the drug transport mechanism. It was observed that the n value is between 0.43 and 0.85, which shows that the release mechanism followed anomalous transport as previously reported by Peppas et al. [35]. Evidence of the transformation of the crystalline drug into an amorphous state and the nanosize range of EZE- loaded NLCs illustrates the enhanced dissolution of EZE. The reason for the variation in rate of release is not fully understood, but it could be due to changes in the surface area, particle size [36]. These results encouraged us to check the bioavailability of the test samples in an animal model.

### 2.9. Accelerated Stability Studies of EZE-NLCs

The effect of temperature was observed before and after the lyophilisation (Virtis-Bench Top Lyophilizer, Spinco Biotech Pvt. Ltd., Chennai, India). Initial particle size, PDI, and drug content were determined at the time of preparation and then the batch was divided into two equal portions, which were stored under different temperature conditions in a refrigerator (2–8 °C) and at room temperature (28–30 °C). Samples were withdrawn for up to 2 months and were subjected to drug content and particle size analysis, PDI, and zeta potential measurements. The results are tabulated in Table 4. At room temperature the formulation remained stable for one week after which aggregation was observed. Thus, EZE-NLCs are less stable at room temperature as compared to cold conditions. The stability study results indicated that the optimized batch was quite stable under refrigerated conditions.

### 2.10. In-Vivo Animal Study

#### 2.10.1. Effect of EZE-NLCs on Lipid Profile in High Fructose Diet (HFD)-Induced Hyperlipidemia

##### Serum Lipid Analysis

HFD is well known for promoting hypercholesterolemia, hypertriglyceridemia and hepatic steatosis [37]. It is evident that serum TC was significantly increased (*p* < 0.001) for HFD-feed rats in comparison with the normal HFD-induced group. TC levels showed a significant decrease from the EZE-NLCs-treated groups (Group IV) in comparison with the HFD rat and EZE-treated groups. Results of TG and LDL levels show a substantial increase compared to normal rats in rats fed with HFD. Significant decreases in TG and LDL levels compared with HFD fed rats and EZE treated rats have been observed following treatment with EZE-NLCs. The highest elevation in the treated EZE-NLC group was observed with regard to HDL values, reflecting a higher bioavailability of the EZE and greater formulation superiority. It is known that HDL has a protective function in CVD because it promotes cholesterol movement in catabolism and excretion from the peripheral tissues to the liver. Table 5 shows the serum lipid profile. The EZE-NLCs were better protected compared to the EZE.

#### 2.10.2. Histopathological Analysis

The most effective method for the acceptable assessment of anti-hyberlipidemic activity in a developed formulation remains liver biopsies and histological testing. Figure 8 exhibits histological photomicrographs of various categories of liver part tissues. Figure 8A shows the normal (group I), of which liver parts display typical liver structures with standard polygonal circular, central vein hepatic cells without cups cell and sinusoids infiltration. On the contrary, the rats with HFD are category II; Figure 8B reveals irregular hepatic tissue morphology, dilation, sinusoidal enlargement, artery swelling, vacuolization, and invasion of Kupffer cells. HFD and thus hypercholesterolemia has been confirmed to be considered a risk factor for hepatic fibrosis as well as atherosclerosis and coronary artery disease. With EZE and EZE-NLCs, Group III and IV demonstrated an increase in hepatocyte structure. Restorations of standard histological liver structures expressed in hepatic fibers, Kupffer cells and vacuoles were accomplished. It is clear that EZE-NLCs decreased the histological changes considerably relative to those found in rats fed with HFD and the rats treated with EZE. It is worth noting that steatohepatitis can be strengthened, a development which can lead to end-stage liver diseases. The lesions histological seen in Figure 8A the yellow arrow represents the narrowing of the vascular and the red arrow shows the enlargement. Table 6 details the scoring for the histopathological lesions.

## 3. Materials and Methods

Ezetimibe was kindly donated by TEVA Pharmaceuticals (Mumbai, India). Capmul PG 8 (propylene glycol monocaprylate, CAS NO. 31565-12-5) was obtained as a gift sample from Abitec Corporation (Columbus, OH, USA) Glyceryl monostearate (CAS NO. 31566-31-1) was obtained as a gift sample from Gattefosse (Mumbai, India). Poloxamer 188 (CAS NO. 9003-11-6) was purchased from Balaji Drugs (Surat, India). All other chemicals and reagents were of analytical grade and obtained from the central laboratory store of the R. C. Patel Institute of Pharmaceutical Education and Research, Shirpur. To optimize the formulation the Design Expert^®^ version 10.0.2 software was used (Minneapolis, MI, USA).

### 3.1. Initial Solubility Studies for screening of Lipids

#### 3.1.1. Solubility of EZE in Liquid Lipids

To investigate the solubility of EZE in liquid lipids, a weight of 15 mg of EZE was mixed with 2 mL of the tested oil within a test-tube, and 3 mL methanol was added until a homogeneous mixture was obtained. The aqueous phase of the above mixture was separated from the lipid by centrifugation at 25,000 rpm for 20 min on a high-speed centrifuge. The clear supernatant obtained after centrifugation was suitably diluted and analyzed by UV spectroscopy (Shimadzu, Tokyo, Japan) at 233 nm λ_max_ for EZE content to study its partitioning behavior with various lipids. The liquid oil that dissolved the highest amount of EZE was the only one selected for use in the test of “miscibility” [38].

#### 3.1.2. Solubility of EZE in Solid Lipid

For studying the solubility in solid lipids, accurately weighed EZE (30 mg) was taken in a test tube; the solid lipid was added in increments of 0.5 g, and the test tube was heated in a controlled temperature water bath kept at 80 °C till the clear melt was achieved. By this way, the quantity of solid lipid required to solubilize 30 mg of EZE was estimated [39,40].

### 3.2. Preparation of Ezetimibe NLCs

EZE-loaded NLC were formulated via hot high-pressure homogenization technique. Briefly, a 70:30 GMS (solid lipid) and Capmul PG 8 (liquid lipid) concentration was kept constant. In the lipidic phase the solid lipid such as GMS was melted, and 10 mg EZE (drug) was utilized. EZE was dissolved in the melted solid lipid phase then liquid lipid (Capmul PG 8) was added to the above melted lipidic phase and further heated to 80 °C to obtain a clear lipid phase. Meanwhile, an aqueous surfactant solution with hydrophilic surfactant (Poloxamer 188) was prepared and heated to the same temperature. The hot surfactant solution (aqueous phase, 100 mL) was then dispersed dropwise in the hot lipid phase to form a milky white pre-emulsion with continuous stirring on a mechanical stirrer (Remi Instruments Ltd., Mumbai, India) at 1500 rpm for 15 min. Then this warm pre-emulsion was introduced into a high pressure homogenizer (PANDA 2K, NiroSoavi, Parma PR, Italy) at optimized 700 bar pressure and seven cycles to form the NLC dispersion. Then the NLCs so formulated were allowed to cool to room temperature and further used for characterization [41,42].

### 3.3. Design of Experiment (DOE)

The design matrix was built by the statistical software package Design Expert^®^ (version 10.0.2). A three factor, two level (2^3^) full factorial design was used to study the effect of independent variables on dependent variables. Three factors, viz. the amount of Poloxamer 188 (A), pressure of HPH (B) and number of cycles (C) were used and coded as −1, 0 and +1 for low, medium and high levels, respectively (Table 7). The particle size (nm) (Y1), %EE (Y2) was taken as the response variables. In this design, experimental trials were performed at all eight possible combinations. All other formulation variables and processing variables were kept invariant throughout the study [43].

The quadratic model equation is as follows:(3)$$y=b_{0}+b_{1}A+b_{2}B+b_{3}C+b_{12}AB+b_{13}AC+b_{23}BC$$
where y is the measured response associated with each factor level combination; $b_{0}$ is an intercept, $b_{1}$, $b_{2}$ and $b_{3}$ are regression coefficients computed from the observed experimental values of y. A, B and C are the coded levels of independent variables, whereas dependent variables were particle size and % EE.

### 3.4. Determination of Particle Size, PDI and Zeta Potential of the EZE-NLCS

The mean particle size was determined by photon correlation spectroscopy (PCS) using a Zetasizer (Nano ZS 90, Malvern Instruments) [42]. This technique is non-invasive to NLCs and efficient to determine the mean particle size and PDI in the submicron region. EZE-NLC formulations were diluted with distilled water to get optimum 50–200 kilo counts per second (kcps) for measurements.

Based on the Helmholtz-Smoluchowski equation, the surface charge of the NLCs was determined by measuring the zeta potential using the same equipment. Zeta potential measurements were run at 25 °C with electric field strength of 23 V/m [44].

### 3.5. Percentage Entrapment Efficiency (%EE)

The amount of EZE entrapped in NLCs was evaluated per a previously reported method [41]. Mathematically it is ratio of the amount of entrapped drug to the amount of total drug used for the preparation of nanoparticles. In brief, 2 mL of the NLCs dispersion was placed in an ultracentrifuge at 10,000 rpm for 30 min at 4 °C [45,46]. The supernatant was removed, suitably diluted with methanol and analyzed spectrophotometrically at 233 nm.
$$\%\mathrm{EE}=\frac{Wt.of\;drug\;used\;in\;formulation\;–Wt.\;of\;unbound\;drug\;in\;supernatant}{Wt.of\;drug\;used\;in\;formulation\;}\times 100$$

### 3.6. Differential Scanning Calorimetry (DSC)

DSC analysis was employed for solid state characterization pure EZE, GMS, a physical mixture of EZE and lyophilized EZE-NLC. Phase transition temperature and energy was verified using differential scanning calorimetry (Mettler Toledo, Columbus, OH, USA). An accurately weighed 2 mg sample was placed in an aluminum pan and sealed with a lid. In the scanning process, a heating rate of 10 °C/min was applied in the temperature range from 30 to 250 °C (except for GMS, because of its low melting point, where the temperature range was kept from 30 to 100 °C) with a nitrogen flow of 5 mL/min [47].

### 3.7. Transmission Electron mMicroscopy (TEM) analysis

The nanostructure of EZE-NLCs was observed using TEM (TECNAI-G2, 200 kV, HR-TEM, FEI, Amsterdam, The Netherlands). A diluted NLC drop was placed on a paraffin sheet and a carbon coated grid was put on the sample and left for 1 min to allow the NLC to adhere on the carbon substrate. Any excess NLC was removed with the help of filter paper. Then the grid was placed on the drop of tungstate phosphor (1%) for 10 s. The remaining solution was removed by absorbing the liquid with a piece of filter paper and samples were air dried and examined by TEM [48]

### 3.8. X-ray Diffraction Measurements

Powder X-ray diffraction (PXRD) was performed to analyze the crystalline or amorphous nature of EZE loaded NLCs. Experiments were performed on a powder X-ray diffractometer (Bruker AXS, D8 Advance, Rosenheim, Bavaria, Germany) using the Cu-Kα line as a source of radiation. The samples were scanned over a 2θ scale at a scan rate of 3°/min. Samples used for the study were pure EZE, a physical mixture of EZE with GMS and lyophilized EZE-NLC.

### 3.9. Determination of In-Vitro Drug Release from EZE-NLCs

The in-vitro drug release from EZE-NLCs (i.e., optimized batch F8) were carried out in 0.45% sodium lauryl sulphate in 0.05 M acetate buffer (pH 4.5) by using the dialysis bag diffusion technique [49]. A dialysis bag (molecular weight cut-off 12–14 kDa) was sealed at one end; 2 mL of NLCs dispersion was poured into it and the other end was then tightly sealed. Then it was placed in a beaker containing 0.05 M acetate buffer at 37 ± 1 °C and magnetically stirred at 50 rpm. At predetermined time intervals up to 24 h, 0.5 mL samples were withdrawn by filtration through 0.22 µm filter (Millipore, Darmstadt, Germany) with the bath conditions were maintained at the same temperature. The filtrate was suitably diluted if necessary and analyzed using a UV spectrophotometer at λ_max_ 233 nm.

### 3.10. Accelerated Stability Studies of EZE-NLCs

For commercialization of NLC products, their stability should be ensured during the shelf life (storage or transport). A pharmaceutically acceptable NLCs demands a shelf life of at least one year, which is a prerequisite criterion to market such nanoformulations. Especially for NLCs, the drug retention capacity and the particle size should be maintained during this storage time. Hence the drug leakage, particle size growth and Zeta-potential are the parameters to be studied [49]. Stability studies for prepared NLCs were carried out for up to 2 months and stability was accessed by drug content measurements, particle size, polydispersity index, and zeta potential of the NLCs. The factor considered for the study was the effect of temperature on storage. The stability study was conducted under three temperature conditions, at the time of preparation, at room temperature (28–30 °C), and under refrigerated conditions (2–8 °C).

### 3.11. In-Vivo Anti-Hyperlipidemic Activity of EZE-NLCs

#### 3.11.1. Animals

The rats (male Wistar-150–180 gm) were procured from the laboratory animal facility of our Institute. They were housed at the standard temperature (22 °C) and relative humidity (65%) conditions established by the ethics committee. All animals were fed with standard pellet chow diet and water ad libitum. The rats were maintained in conformity with the regulations laid down by the Committee for the Purpose of Control and Supervision of the Experiments on Animals (CPCSEA), Government of India. The experimental protocol was approved by the Institutional Animal Ethics Committee (IAEC) of R. C. Patel Institute of Pharmaceutical Education and Research, Shirpur, District-Dhule, Maharashtra, India.

#### 3.11.2. Induction of Hyperlipidemia

Hyperlipidaemia was achieved by oral feeding of the hypocholesterolemic inductor high fat diet in rats. It consists of protein 20% Kcal, fat 45 % Kcal, 35% Kcal, carbohydrate 53% Kcal, energy density 4.7% Kcal (HFD, Cat# D12451, Research Diets, Inc., New Brunswick, NJ, USA) for a period of 10 weeks. The signs of hyperlipidemia were confirmed by evaluating the lipid content at the end of the 6th week. The rats having hyperlipidemia were included in the study. After six weeks’ treatment was started and at the end of the study or during last treatment schedules different parameters were evaluated [50].

#### 3.11.3. Experimental Design

Group-I (Control group): received only vehicle (1 mL/kg p.o.) and normal diet

Group-II (HFD group): received HFD and vehicle for 10 weeks

Group-III (HFD + EZE): received HFD for 10 weeks and ezetimibe (10 mg/kg p.o.) from 6th to 10th week

Group-IV (HFD + EZE-NLCs): received HFD for 10 weeks and EZE-NLCs (06 mg/kg p.o) from 6th to 10th week

At the end of the experimental protocol and after overnight fasting, blood samples were collected by retro-orbital plexus puncture under mild anesthesia. Serum was separated by centrifugation at 3000 rpm for 15 min for the assessment of lipid profile and other parameters like total cholesterol (TC), triglycerides (TG), high density lipoprotein cholesterol (HDL-c), and low density lipoprotein (LDL-c) [51].

#### 3.11.4. Evaluation of Serum Lipid Profiles

TC, TG and total lipids including LDL-c, HDL-c were evaluated using in vitro diagnostic kits according to the manufacturer’s instructions (ERBA Diagnostics, ALPCO, Salem, NH, USA). Low density lipoprotein cholesterol (LDL-C) level was calculated by using Friedewald’s equation (*n* = 6).

#### 3.11.5. Histopathology

The rats (*n* = 4) were sacrificed at the end of last treatment and the liver was excised, weighed, washed with the saline solution. The liver tissues were fixed in 10% formaldehyde and then embedded in paraffin wax, which was sectioned (3–4 mm thick), mounted on glass slides and stained using hematoxylin and eosin. The stained slides were observed for histopathological changes under light microscope (CM-1100, Leica, Wetzlar, Germany). The five fields from each sample were analyzed and scored for liver including vacuolization of hepatocytes, enlargement of sinusoids, Kupffer cell infiltration, vascular congestion was indicated in Table 5. The observations were scored as severe, moderate, mild and nil.

#### 3.11.6. Statistical Analysis

The data were expressed as mean±SEM and analyzed by one/two-way analysis of variance (ANOVA)followed by Dunnett’s post hoc test using Graph-Pad Prism 6 software (Graph Pad Software, La Jolla, CA, USA).

## 4. Conclusions

NLC formulations have been explored for their numerous advantages like solubility enhancement and enhanced delivery efficiency. In the present study we have successfully developed EZE-loaded NLCs via a high pressure homogenization technique. The optimization to get the suitable processing variables was carried out through a 2^3^ full factorial design. TEM studies revealed the spherical shape of EZE-NLC with perfect boundaries and DSC studies manifested the absence of crystalline state of the EZE and displayed the drug excipients’ compatibility. Moreover, in-vitro studies demonstrated the sustained release profile of developed EZE-NLCs over the period of time which displayed anomalous transport. An accelerated stability study showed no significant change in drug content and particle size, consequently a stable formulation was achieved. The NLCs due to their size and lipophilic characteristics may help avoid first pass metabolism. The EZE-NLCs displayed increased efficiency which may offer a better control over the disease dealing with respect to reduction in dose and dose-related side effects. EZE-NLCs have the potential to lower hyperlipidemia, as evident from the serum lipid profiles. Furthermore, treatment with EZE-NLCs could meaningfully restore the liver tissue histology after intoxication with cholesterol.

We, therefore conclude that NLCs can be suitable approach for improving the oral bioavailability and pharmacological bioactivity of the poorly water-soluble drugs. This improved bioavailability in case of the EZE-NLCs could be attributed to many factors including nanoscale particle size, change of crystalline drug to amorphous and improved solubilization of the drug. However, the full significance of EZE-NLCs may only be established when evaluated clinically.

## Figures and Tables

**Figure 1 molecules-26-01485-f001:**
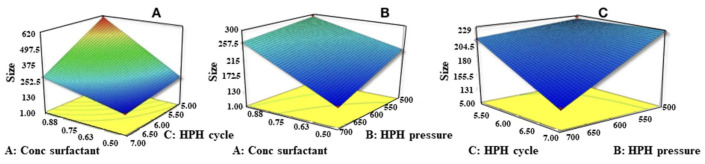
Response surface plots of particle size (Z-Avg) versus conc. of surfactant, number of HPH cycles and pressure at HPH. (**A**) Effect of conc. of surfactant and number of HPH Cycles on particle size, (**B**) effect of conc. of surfactant and pressure at HPH on Particle size, (**C**) effect of pressure at HPH and number of HPH Cycles on particle size.

**Figure 2 molecules-26-01485-f002:**
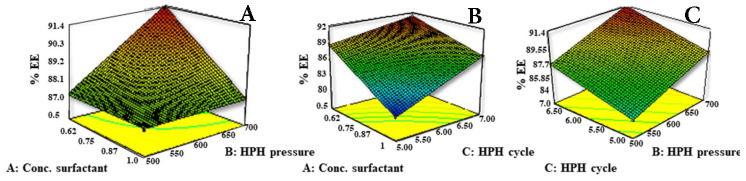
Response surface plots of % entrapment efficiency (%EE) versus conc. of surfactant, number of HPH Cycles and pressure at HPH, (**A**) Effect of conc. of surfactant and pressure at HPH on % EE, (**B**) effect of conc. of surfactant and number of HPH cycles on % EE, (**C**) effect of pressure at HPH and number of HPH Cycles on % EE.

**Figure 3 molecules-26-01485-f003:**
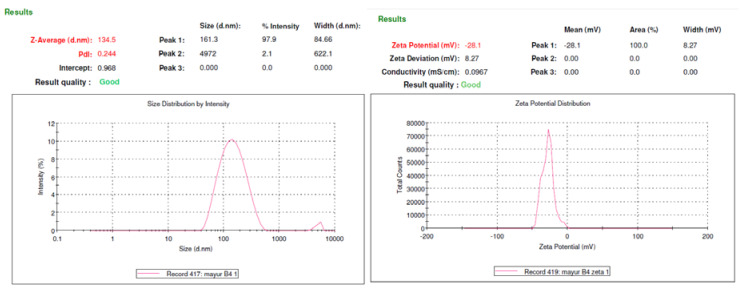
(**A**) Particle-size of EZE-loaded NLCs (F8 Optimized batch) (**B**) Zeta- Potential of EZE-loaded NLCs (F8 Optimized batch).

**Figure 4 molecules-26-01485-f004:**
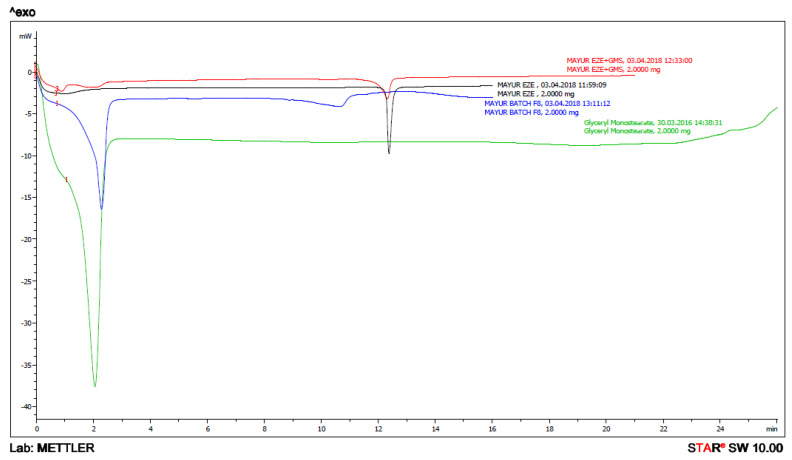
DSC thermogram overlay of (A) physical mixture of ezetimibe and glyceryl monostearate; (B) ezetimibe; (C) ezetimibe-loaded NLC formulation (D) glyceryl monostearate.

**Figure 5 molecules-26-01485-f005:**
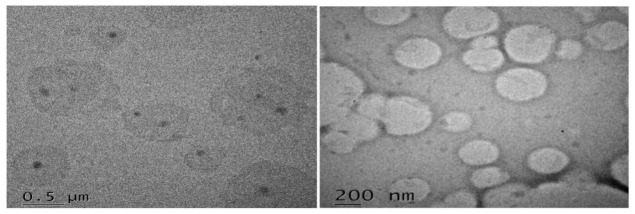
Morphology of EZE-loaded NLC particles visualized by transmission electron microscopy (TEM) voltage of 60–80 kV (magnification ×10,000).

**Figure 6 molecules-26-01485-f006:**
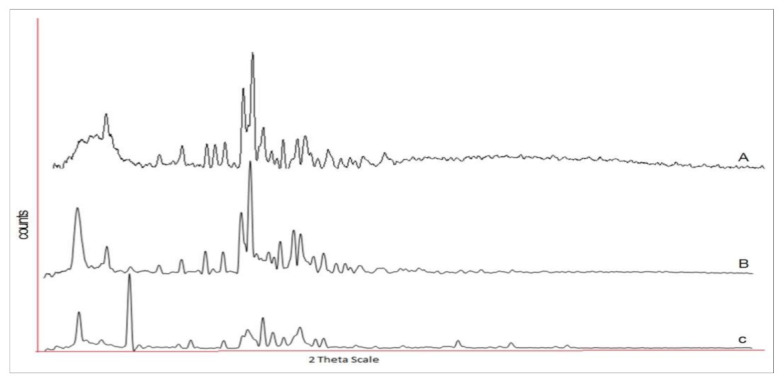
X-ray diffractograms of (A) ezetimibe (B) physical mixture of ezetimibe and glyceryl monostearate (C) ezetimibe-loaded NLC formulation.

**Figure 7 molecules-26-01485-f007:**
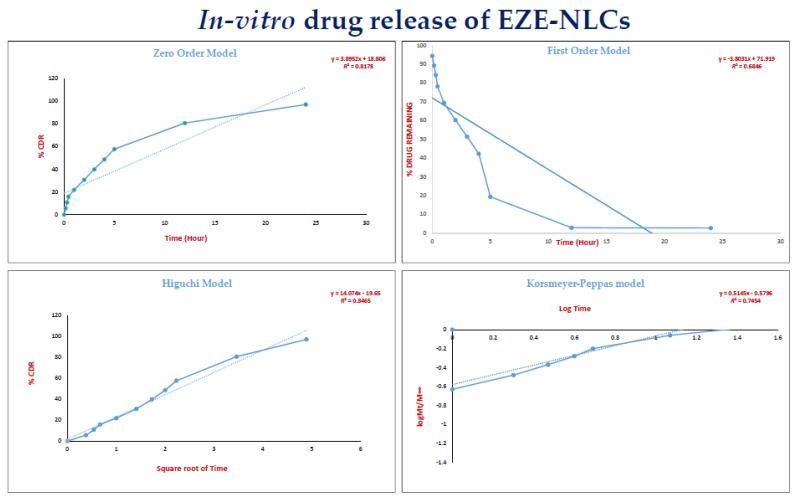
The release data of EZE-NLCs demostrating zero order, first-order, Highuchi’s and the Korsmeyer-Peppas model.

**Figure 8 molecules-26-01485-f008:**
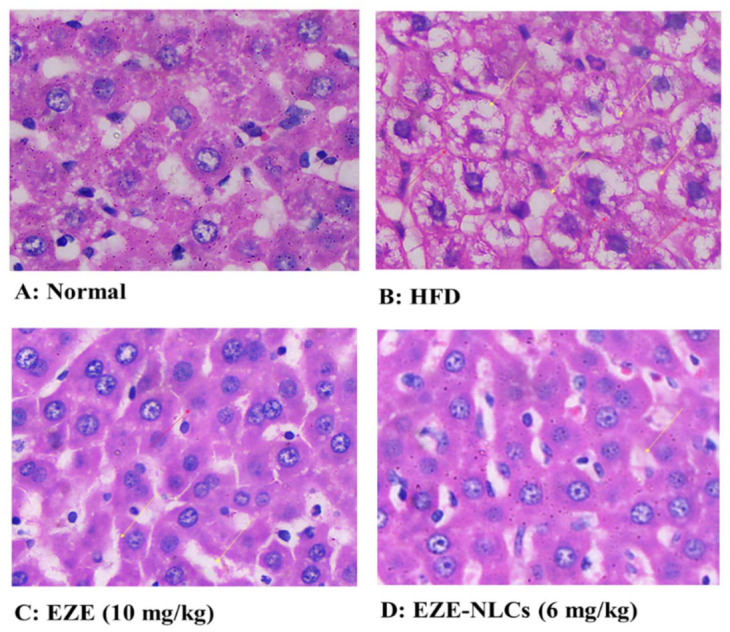
Effect of EZE-NLCs on histopathological changes (10×). (**A**) shows the normal liver section exhibits regular liver architecture with normal polygonal hepatic cells; (**B**) HFD shows abnormal architectural organization of hepatic tissue, dilatation, enlargement of sinusoids, vascular congestion, vacuolization, and Kupffer cell infiltration; (**C**) and (**D**) shows improvements of hepatocyte structures, Kupffer cells and with without vacuoles was attained. It is clear that EZE-NLCs remarkably reduced the histological changes when compared to that observed in HFD fed rats and the EZE treated rats.

**Table 1 molecules-26-01485-t001:** Formulation and evaluation of ezetimibe-loaded NLCs by 2^3^ full factorial design: Factors, their levels and transformed values and responses.

Batch	Particle Size (nm)	PDI	Ζ Potential (mV)	Factor 1	Factor 2	Factor 3	Response 1	Response2
				A Conc. Surfactant %	B HPHP Ressure Bar	C No. of HPH Cycle	Particle Size (nm)	Entrapment Efficiency %
F1	614.4 ± 3.3	0.798 ± 0.02	−22.3 ± 0.5	1.00	700.00	5.00	614.4 ± 3.3	80.09 ± 1.9
F2	550.1 ± 6.0	0.715 ± 0.04	−19.1±0.6	1.00	500.00	5.00	550.1 ± 6.0	80.40 ± 0.5
F3	208.0 ± 2.9	0.273 ± 0.03	−23.6 ± 0.4	0.50	700.00	5.00	208.0 ± 2.9	88.51 ± 2.4
F4	262.0 ± 4.9	0.524 ± 0.02	−27.8 ± 0.6	1.00	700.00	7.00	262.0 ± 4.9	87.22 ± 0.7
F5	227.2 ± 0.9	0.358 ± 0.02	−20.1 ± 0.9	0.50	500.00	5.00	227.2 ± 0.9	84.07 ± 3.0
F6	225.5 ± 4.8	0.282 ± 0.02	−21.9 ± 0.5	0.50	500.00	7.00	225.5 ± 4.8	87.12 ± 2.0
F7	296.7 ± 2.7	0.619 ± 0.07	−26.3 ± 0.5	1.00	500.00	7.00	296.7 ± 2.7	87.04 ± 1.9
F8	134.5 ± 2.5	0.244 ± 0.03	−28.1 ± 0.3	0.50	700.00	7.00	134.5 ± 2.5	91.32 ± 1.8

Mean ± SD (*n* = 3).

**Table 2 molecules-26-01485-t002:** Summary of results of regression analysis for responses.

Response	Model	*R* ^2^	Adjusted *R*^2^	Predicted *R*^2^	F value	P Value	SD (±)	% CV	Mean
**Particle Size**	2F1	0.9996	0.9969	0.9717	53.506	0.0394	9.59	3.05	314.81
**Entrapment Efficiency%**	2F1	0.9996	0.9975	0.9772	60.906	0.0354	0.20	0.23	85.68

**Table 3 molecules-26-01485-t003:** In vitro release kinetic parameter of model fitting of optimized batch of EZE-NLCs.

Model	(*R*^2^) Coefficient of Determination
Zero order	0.8178
First order	0.6846
Highuchi’s	0.8465
Korsmeyer-Peppas	0.7454

**Table 4 molecules-26-01485-t004:** Stability study of ezetimibe-loaded NLCs during storage at different temperature.

	PS (nm)	PDI	EE %
At the time of preparation	134.5 ± 2.5	0.244 ± 0.03	91.32 ± 1.8
**Storage at Room Temperature (28–30 °C)**			
After 1 Month	197.1 ± 4.9	0.333 ± 0.02	88.4 ± 1.5
After 2 Month	282.3 ± 2.0	0.417 ± 0.04	89.7 ± 1.5
**Storage Condition in Refrigerator (2–8 °C)**			
After 1 Month	182.3 ± 3.0	0.329 ± 0.02	90.1 ± 2.9
After 2 Month	214.6 ± 3.6	0.382 ± 0.01	90.6 ± 1.4

Mean ± SD (*n* = 3).

**Table 5 molecules-26-01485-t005:** Effect of EZE-NLCs, on serum level of total cholesterol, triglyceride, high-density lipoprotein and low-density lipoprotein.

Group	Triglyceride (mg/dL)	Total Cholesterol (mg/dL)	LDL-c (mg/dL)	HDL-c (mg/dL)
Normal	76.26 ± 1.96	112.2 ± 4.02	36.40 ± 4.60	84.84 ± 4.37
HFD	193.00 ± 7.23 ^###^	220.7 ± 12.51 ^###^	75.95 ± 4.58 ^###^	41.19 ± 2.72 ^###^
HFD + EZE (10 mg/kg)	110.3 ± 15.47	144.4 ± 4.61	51.58 ± 6.65	62.70 ± 6.10
HFD + EZE-NLCs (6 mg/kg)	90.87 ± 4.92 ^***^	119.0 ± 8.28 ^***^	35.99 ± 3.63 ^***^	82.99 ± 3.19 ^***^

Data was represented as mean ±SEM, analysed by one-way analysis of variance followed by Bonferroni’s post hoc test. ^###^
*p* < 0.001 as compared with normal and *** *p* < 0.001 as compared with HFD. *p* < 0.05 is considered as statistically significant.

**Table 6 molecules-26-01485-t006:** Effect of EZE-NLCs on histopathological changes, (−): Nil; (+): mild; (++): Moderate; (+++): Severe.

	Vacuolization of Hepatocytes	Enlargement Sinusoids	Kupffer-Cell Infiltration	Vascular Congestion
Normal	−	−	+	−
HFD	+++	++	+++	+++
HFD + EZE (10 mg/kg)	++	+	+	+
HFD + EZE-NLCs (6 mg/kg)	−	+	−	−

**Table 7 molecules-26-01485-t007:** Factors, their levels and transformed values in formulation of ezetimibe-loaded NLCs by a 2^3^ full factorial design: variables and their constraints in Design Expert^®^ 10.0.2.

Independent Variables	Level		
	Low (−1)	Medium (0)	High (+1)
A: Amount of Poloxamer	0.5	0.75	1
B: Pressure of HPH	500	600	700
C: Number of cycles	5	6	7

## Data Availability

All data are already provided in the manuscript.

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
