# Peer review of "Ezetimibe-Loaded Nanostructured Lipid Carrier Based Formulation Ameliorates Hyperlipidaemia in an Experimental Model of High Fat Diet"

_molecules, 2021, doi:10.3390/molecules26051485_

Round 1
Reviewer 1 Report
The manuscript is dealing with the application of nanotechnology for a better availability of drug, that in the specific is EZE, as an anti-hypercholesterolemic compound. The data presented are encouraging; however, the work need some improving. In particular the morphological investigation should be better performed. The histopathology of liver is too superficial. The TEM picture of EZE loaded NCL is showing a heterogeneity of size and many particles are fusing together. This is not described neither discuss. The data of DLS (graphs for example) should be presented)
In the legend of figure 6, it is reported that vacuoles are present or not in the Kupffer cells. The magnification impede to verify the statement. No explanation for the arrows of different colours is given.
Reviewer 2 Report
In this study ezetimibe (EZE), a drug with low solubility used to treat hypercholesterolemia, was encapsulated into nanostructured lipid carriers (NLC) via high pressure homogenization technique. The optimum formulation was selected employing a three factor, two level (23) full factorial design and characterized with various physicochemical techniques. Finally, the efficiency of the optimum EZE formulation was studied in vivo.
In my opinion, this manuscript lacks of novelty since there are many works in the literature describing studies with ezetimibe encapsulated into NLC. Moreover, in the publication: Pranav Shah et al. Formulation and Evaluation of Ezetimibe Loaded Solid Lipid Nanoparticles. Indo American Journal of Pharmaceutical Research.2017:7(08), EZE was encapsulated into the same nanostructured lipid carriers and analogous study was presented. Additional comments are as followed.
- The novelty of the study should be clearly presented.
- The experimental part should be better present, providing full experimental details. Moreover, in the experimental part: The selected solid lipid and liquid lipid as well as their weight ratio should be mentioned from the beginning (section 3.1). In section 3.2, what is the concentration of the EZE loaded dispersion that was used. In section 3.6, DSC analysis was not employed to study the morphology of the nanoparticles. Please correct it. In section 3.9: Initially the authors said that the release medium is 0.45% SLS in 0.05 M Acetate buffer and after some lines they said that they used 200 ml PBS media.
- The discussion of the results and the conclusion are not clearly presented. These sections should be rewritten focusing on the necessity of the optimization of the EZE-NLC preparation and advantages of this system. Additionally, a comparison to other studies or commercially available product is missing. In the conclusion, the authors stated that “The EZE-NLCs offers a better control over the disease dealing with bioavailability, dosing frequency, and dose related side effects.” This cannot conclude from their results since the presented animal experiments related only with EZE-NLC efficiency.
- Table 2 and 3 should be merged.
- The authors should carefully polish the English writing throughout the manuscript.
- Please correct some typo-errors, e.g. correct “ξ-potential” to “ζ-potential” or “23 full factorial design” to “23 full factorial design”.
- Explain the abbreviations presented in the manuscript, such as GMS, SLS, etc.
Author Response
"Please see the attachment."

Reviewer 3 Report
- The Introduction section needs improvement. References to similar studies have to be included and a comparative evaluation of published results has to be made.
- According to the Authors Guidelines of Molecules, the Materials and Methods section has to be placed after the Results and Discussion Section.
- Sections Material and Methods have to be united in one Section. Besides, all chemicals and reagents used in the study have to described I details including the CAS No., concentrations, purity, supplier.
- The appliances used, e.g. UV-Vis spectrophotometer – Model, type, supplier, have to be added.
- The information given in section 1. Initial solubility studies for screening of lipids is unclear. The section has to be revised.
- All abbreviations used in the manuscript have to be written in full when first mentioned – e.g PDI, GMS!
- The numbers of the Tables in the text of the manuscript have to correspond to those of the table captions! All Tables have to be cited in the text in the order they appear in the manuscript!
- Tables 1 and 2 present experimental data, thus they have to be placed in Section Results, but not in Materials and Methods.
- The wavelengths stated are different – 233 or 234 nm?
- Equation (1) – the subscripts have to be written correctly!
- Table 2: Probably the ± values represent the standard deviation; however, it is not clear and have to be specified! The units of the parameters are missing!
- It is not clear if fresh PBS was added after the withdrawal of 5 mL samples during the in vitro experiments! As the concentration will change additionally and the results will not be correct.
- The need and aim for the application of a three factor, two level (23) full factorial design are not clearly described. The authors have to specify the necessity and the goal of the analysis!
- In the in vivo studies: what was the number of animals in each the 4 group?
- This part is totally unclear: “For the preparation of EZE-NLCs, GMS was selected as a solid matrix and Capmul PG 8 as liquid lipid with constant concentration 70:30 respectively, as optimized in partition behavior study. Thus, depending upon the result obtained from trial batches, the optimal value range of the variables was surfactant concentration (X1) of 0.5 – 1gm…” The authors have to explain the
- Section 4.1. Partitioning behavior of EZE between the lipids – the information presented is not scientifically proven.
- The aim of the application of the factorial analyses is totally unclear, as mentioned above. Besides, the information presented in section 2. Development and optimization of EZE loaded NLCs does not present any scientifically significant results. The discussion is totally missing. What is the sense???
- Sections 4.2 and 4.3 have to be placed after section 4.4, as in a scientific publication a logical order has to be followed. The authors apply models before they have presented and discussed the experimental data!
- The authors have to supply a figure with higher quality as Figure 4.
- Section 4.8. has to be completely revised. The experimental data presented in Figure 5 D do not correspond to the conclusion for the applicability of the Higuchi model! Obviously, these data are approximated with a linear function which is totally unapplicable in this case. Detailed mathematical modelling of the experimental in vitro results is necessary! Besides the quality of Figure 5D has to be improved significantly.
- The Discussion of the results is not appropriate for a scientific paper of high quality. It has to be revised.
- The References in the References list have to be written in accordance to the Author Guidelines of the journal.
- The text after Author contributions has to be removed as it is not part of the present manuscript but from the template.
In conclusion, the idea and goal of the manuscript cover the investigations conducted. However, significantly more detailed analyses and discussions are needed. The experimental data have to presented in a more detailed and clear way. The manuscript could even be divided into two separate papers: in vitro and in vivo investigations. The manuscript needs significant revision in view of grammar and style (see the highlighted parts in the pdf file). There are plenty of technical errors, as well.
Author Response
"Please see the attachment"

Round 2
Reviewer 1 Report
The Authors have taken into consideration all the comments raised. The manuscript can be accepted fo publication
Author Response
We are really thankful to you for your valuable comments and thoughtful suggestions which helped us to improve and rectify our manuscript.
We make sure that we have revised the complete manuscript as per suggestion.
"Please see the attachment"

Reviewer 2 Report
The authors were taken under consideration all my suggestion and now I think that the manuscript is suitable for publication.
Author Response
We are really thankful to you for your valuable comments and thoughtful suggestions which helped us to improve and rectify our manuscript.
We make sure that we have revised the complete manuscript as per suggestion.
Thank you once again!
"Please see the attachment"

Reviewer 3 Report
The authors have complied with the remarks of the reviewer, so the manuscript was substantially improved. However, the following weak points should also be considered:
- The authors replied that the number of experimental animals in the studied groups are 4 in each group, which is insufficient for the statistical significance of the experimental results. The number of experimental animals have to minimum 5 or 6.
- The quality of the plots in Figure 7 have to be improved. The authors have to remove the title within the plots and increase the font size.
- Why did the authors choose exactly the Hixtson-Crowell model? A comparative analyses between a few release models have to be accomplished so that the most adequate between them could be selected for the experimental data. The model parameters and error functions have to presented in a separate table. What type of analysis was applied: linear, non-linear regression analyses?
- The are a number of spelling mistakes, e.g. the name of the mathematical model.
Author Response
"Please see the attachment"

This manuscript is a resubmission of an earlier submission. The following is a list of the peer review reports and author responses from that submission.